# Ecosystem-wide metagenomic binning enables prediction of ecological niches from genomes

Johannes Alneberg[1], Christin Bennke[2], Sara Beier[2,3], Carina Bunse[4,5,6], Christopher Quince[7], Karolina Ininbergs[8,10], Lasse Riemann[9], Martin Ekman[8], Klaus Jürgens[2], Matthias Labrenz[2], Jarone Pinhassi[4] & Anders F. Andersson[1✉]

The genome encodes the metabolic and functional capabilities of an organism and should be a major determinant of its ecological niche. Yet, it is unknown if the niche can be predicted directly from the genome. Here, we conduct metagenomic binning on 123 water samples spanning major environmental gradients of the Baltic Sea. The resulting 1961 metagenome-assembled genomes represent 352 species-level clusters that correspond to 1/3 of the metagenome sequences of the prokaryotic size-fraction. By using machine-learning, the placement of a genome cluster along various niche gradients (salinity level, depth, size-fraction) could be predicted based solely on its functional genes. The same approach predicted the genomes' placement in a virtual niche-space that captures the highest variation in distribution patterns. The predictions generally outperformed those inferred from phylogenetic information. Our study demonstrates a strong link between genome and ecological niche and provides a conceptual framework for predictive ecology based on genomic data.

[1] Department of Gene Technology, Science for Life Laboratory, School of Engineering Sciences in Chemistry, Biotechnology and Health, KTH Royal Institute of Technology, Stockholm, Sweden. [2] Leibniz Institute for Baltic Sea Research, Warnemünde, Germany. [3] CNRS, Laboratoire d'Océanographie Microbienne, LOMIC, Sorbonne Université, Banyuls/mer, France. [4] Centre for Ecology and Evolution in Microbial Model Systems, Linnaeus, University, Kalmar, Sweden. [5] Helmholtz Institute for Functional Marine Biodiversity at the University of Oldenburg (HIFMB), Oldenburg, Germany. [6] Alfred-Wegener-Institut Helmholtz-Zentrum für Polar- und Meeresforschung, Bremerhaven, Germany. [7] Warwick Medical School, University of Warwick, Coventry, UK. [8] Department of Ecology, Environment and Plant Sciences, Stockholm University, Stockholm, Sweden. [9] Department of Biology, Marine Biological Section, University of Copenhagen, Helsingør, Denmark. [10] Present address: Department of Laboratory Medicine, Karolinska Institute, Stockholm, Sweden. ✉email: anders.andersson@scilifelab.se

The ecological niche, as defined by Hutchinson[1], is an n-dimensional space where the dimensions are environmental conditions and resources that define the requirements of a species to persist. Studies on community assembly have shown that species composition is not independent of phylogeny; a phenomenon commonly observed in both macro- and microorganism communities is phylogenetic clustering[2,3], i.e. that the species of a community are more closely related than expected by chance. Likewise, a correlation between phylogenetic relatedness and ecological similarity has been demonstrated for both macro- and microorganisms[4,5]. A natural explanation for these observations is that closely related species encode similar sets of genes (trait conservation), and hence are equipped to survive and reproduce under similar conditions (environmental filtering)[6,7]. Consequently, the genome should define the fundamental niche of an organism, and in conjunction with abiotic and biotic factors, be a strong predictor of its realised ecological niche.

For prokaryotes, where a large number of genomes are available, computational methods have been developed that can infer phenotypes of varying complexity directly from the genome. Thus, not only the proteome[8] and the metabolome[9] can be predicted, but also specific traits[10,11] such as if the organism thrives under oxic or anoxic conditions[12], what substrates it utilises, what temperature range it prefers[13], if it is pathogenic, if it is resistant to specific antibiotics and if it is oligotrophic or copiotrophic[14]. However, it remains to be shown that the distribution pattern of an organism, which reflects its ecological niche, can be predicted directly from the genome. This would be an important step towards building species distribution models that integrate genetic and environmental information, which would potentially lead to models with increased accuracy. The prerequisites for modeling species distributions based on genomic data would be the availability of a large number of genomes from within an ecosystem, together with quantitative data on the abundances of the corresponding organisms across various niche-gradients in the system.

Microorganisms play key roles in marine and freshwater ecosystems by driving the biogeochemical cycles and by forming the base of the food web[15]. Sequencing-based approaches have contributed fundamentally to the understanding of aquatic ecosystems by informing us on how ecosystem functions are distributed across time, space and taxa[16,17]. Shotgun metagenomics offers extensive cataloguing of metabolic and functional capabilities of communities, and combined with genome binning algorithms ecosystem processes can be linked to individual populations[18]. This circumvents the need for cultivation, which is important since only a small fraction of aquatic microorganisms can be readily isolated. Large-scale metagenomic binning has been conducted on samples spanning the global ocean[19,20] and on a collection of temperate lake samples[21]. We recently reconstructed a set of genomes from the Baltic Sea, one of the world's largest brackish ecosystems, and showed that a global brackish microbiome exists with bacterioplankton that are closely related to but genetically distinct from their freshwater and marine relatives[22]. In this study we have conducted large-scale metagenomic binning to obtain an extensive catalogue of microbial genomes sampled across the Baltic Sea in space and time. We show that we can predict the placement of these genomes along principal niche gradients of the ecosystem based solely on what genes they encode.

## Results

### A catalogue of Baltic Sea bacterioplankton genomes.
We conducted genome binning on 123 metagenome samples from the Baltic Sea, a semi-enclosed sea with several established environmental gradients[23]. Most pronounced are the horizontal salinity gradient, extending from near-freshwater conditions in the north to marine conditions in the southwest, and the vertical oxygen gradient, with oxygenated surface water and sub- or anoxic deep waters over extended areas. Microbial communities of the Baltic Sea are known to be highly structured along these gradients[24–26] and also to display pronounced seasonal dynamics[5,27]. Our samples cover variation in geography, depth, season and size fraction, being mainly comprised of samples collected during two trans-Baltic cruises and from time series samplings at two stations (the Linnaeus Microbial Observatory [LMO] and the Askö station) (Fig. 1a).

Each metagenome sample was assembled and binned individually, but using abundance information from across all samples for the binning. Genome binning on this large sample set was facilitated by using Kallisto for contig quantifications[28]. Kallisto, originally developed for RNA-seq quantification, only requires a fraction of the time necessary for exact read-alignment methods while producing quantifications highly correlated to those (Pearson $r = 0.95$; Supplementary Fig. 1). Furthermore, a highly parallel and improved implementation of the binning algorithm CONCOCT[29] was used. Bins that passed quality control were considered metagenome-assembled genomes (MAGs), using ≥75% completeness and ≤5% contamination as criteria[30]. This generated 1,961 MAGs with an average estimated completeness and contamination of 90.9% and 2.5%, respectively. Additional evaluation of the binning procedure was facilitated by an internal standards genome of an organism not expected to be present in this environment (the hyperthermophile *Thermus thermophilus*) which was added to a subset of the samples prior to sequencing. A MAG representing this genome was obtained from 28 of the 29 samples to where it had been added, verifying the sensitivity of the assembly and binning method used (Supplementary Table 1). Together, the MAGs recruited on average 32% of the samples' shotgun reads using 97% nucleotide identity as threshold (Fig. 1b). Excluding samples from the largest (3.0 μm) and smallest (<0.1 μm) size fractions, containing mainly eukaryotic cells and viruses, respectively, increased the recruited proportion to 36%. This is substantially higher than in a recent study based on the Tara Oceans dataset, where 6.8% of the reads could be mapped to the reconstructed MAGs[19]. Thus, the reconstructed genomes represent a large fraction of the planktonic prokaryotes in the Baltic Sea and will provide an important resource for future studies on brackish ecosystems. It also provides an unprecedented opportunity to investigate links between genome and ecosystem.

Since each sample was assembled and binned individually, several MAGs may represent the same species, and the MAGs were therefore clustered based on sequence identity at an approximate species level of 96.5% average nucleotide identity (ANI)[31]. The distribution of ANI values between MAGs confirmed clustering at this level to be appropriate, with a large number of MAG pairs with ANI > 97% but a sharp drop below this point (Fig. 1c). Accordingly, the 1961 MAGs found here, together with 83 MAGs that we previously recovered from one year of seasonal data from station LMO (representing 30 clusters, of which 27 were rediscovered here)[22], formed a total of 355 Baltic Sea clusters (BACLs). Plotting the number of obtained BACLs as a function of number of samples indicates that additional BACLs remain to be detected, although the curve has started to plateau (Fig. 1c).

Phylogenomic analysis of the MAGs using the Genome Taxonomy Database (GTDB)[32] showed that the obtained MAGs were widely taxonomically distributed (Table 1, Supplementary Fig. 2 and Supplementary Data 1), indicating a low phylogenetic bias of the binning method. The largest number of MAGs were recovered from Actinobacteria, Bacteroidetes, Cyanobacteria,

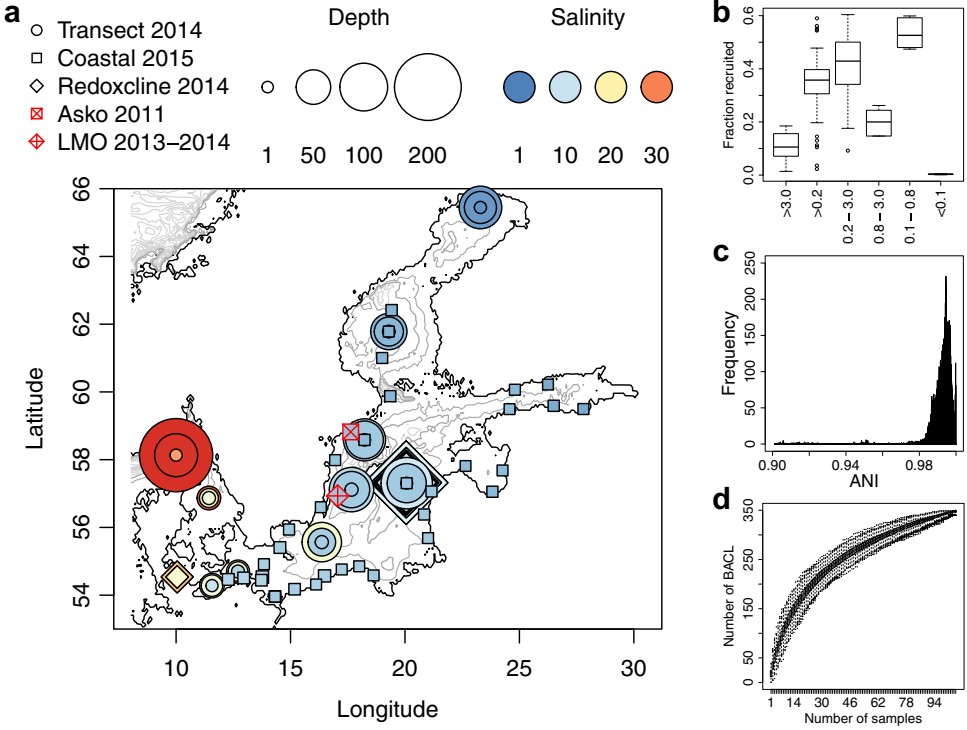

**Fig. 1 Sampling stations and summary of metagenome binning results. a** Map of sampling locations. The included sample sets are indicated with different symbols. The marker colour indicates the salinity of the water sample while the size indicates the sampling depth. The contour lines indicate depth with 50 m intervals. Three of the sample sets have previously been published: *Askö Time Series 2011*[60] ($n = 24$), *Redoxcline 2014*[33] ($n = 14$) and *Transect 2014*[33] ($n = 30$); and two are released with this paper: *LMO Time Series 2013–2014* ($n = 22$) and *Coastal Transect 2015* ($n = 34$). The map was generated with the marmap *R* package[77] using the ETOPO1 database hosted by NOAA[78]. **b** Proportion of metagenome reads recruited to the metagenome-assembled genomes (MAGs), summarized with one boxplot per filter size fraction. **c** Distribution of pairwise inter-MAG distances. Only average nucleotide identity (ANI) values >0.9 are shown. Minimum and maximum within-cluster identity for multi MAG Baltic Sea clusters (BACL) were 96.8% and 100.0%, respectively. Only four BACLs had any MAG with >96.5% identity to any MAG in another BACL. **d** Rarefaction curve showing number of obtained BACLs as a function of number of samples. Boxplots show distributions from 1000 random samplings.

**Table 1 Taxonomic distribution of MAGs.**

| Phylum | Class | Order | Family | Genus | Species | BACL | MAG |
|---|---|---|---|---|---|---|---|
| *Bacteria* | | | | | | | |
| Actinobacteria | 3 | 8 | 14 | 24 | 34 | 68 | 405 |
| Bacteroidetes | 2 | 8 | 18 | 34 | 41 | 87 | 524 |
| Chloroflexi | 3 | 3 | 3 | 3 | 3 | 5 | 12 |
| Cyanobacteria | 2 | 4 | 5 | 8 | 9 | 16 | 66 |
| Desulfobacteraeota | 1 | 1 | 1 | 1 | 1 | 1 | 1 |
| Eisenbacteria | 1 | 1 | 1 | 1 | 1 | 1 | 1 |
| Epsilonbacteraeota | 1 | 1 | 1 | 1 | 1 | 2 | 3 |
| Firmicutes | 1 | 2 | 2 | 2 | 2 | 3 | 9 |
| Gemmatimonadetes | 1 | 1 | 1 | 1 | 1 | 1 | 3 |
| Marinimicrobia | 2 | 2 | 2 | 2 | 2 | 2 | 2 |
| Myxococcaeota | 1 | 1 | 1 | 1 | 1 | 1 | 1 |
| Nitrospinae | 1 | 1 | 1 | 2 | 2 | 2 | 11 |
| Oligoflexaeota | 1 | 1 | 1 | 1 | 1 | 1 | 9 |
| Planctomycetes | 4 | 6 | 9 | 10 | 10 | 28 | 155 |
| Proteobacteria | 2 | 20 | 34 | 57 | 61 | 101 | 612 |
| SAR324 | 1 | 1 | 1 | 1 | 1 | 1 | 1 |
| Verrucomicrobia | 2 | 7 | 11 | 14 | 14 | 25 | 101 |
| Unclassified Bacteria | 1 | 1 | 1 | 1 | 1 | 4 | 10 |
| *Archaea* | | | | | | | |
| Crenarchaeota | 1 | 1 | 1 | 1 | 2 | 2 | 23 |
| Nanoarchaeota | 1 | 1 | 1 | 1 | 1 | 1 | 1 |
| Thermoplasmataeota | 1 | 1 | 1 | 1 | 1 | 2 | 11 |
| Total | 33 | 72 | 110 | 167 | 190 | 354 | 1961 |

Number of unique taxonomic entities assigned at the respective levels. Not all MAGs have obtained a taxonomic classification down to the species level, counts for these are based on the most detailed level for which they have been assigned at.

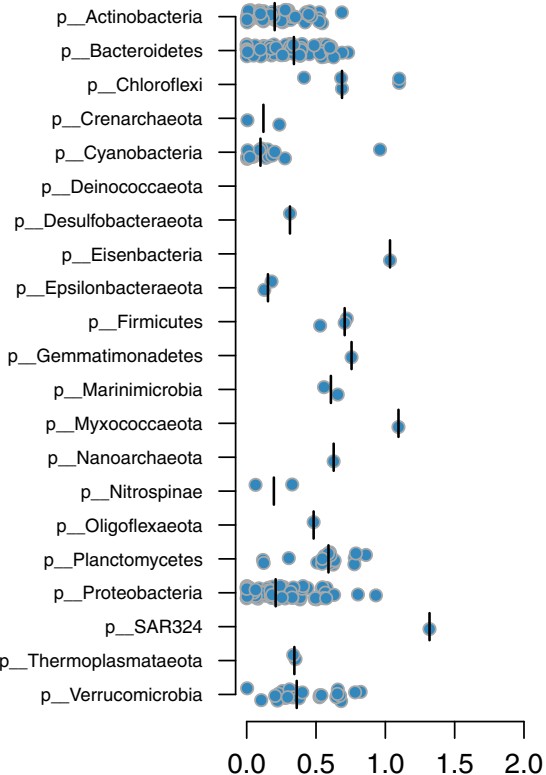

**Fig. 2 Phylogenetic distances between BACLs and nearest GTDB neighbors.** Each circle is a BACL represented by a MAG and the placement along the x-axis indicates phylogenetic distance to the nearest reference genome in the GTDB tree. Distributions are plotted separately for each phylum, with median values indicated by verticallines.

Planctomycetes, Proteobacteria (mainly Alpha- and Gammaproteobacteria) and Verrucomicrobia. This is consistent with previous marker gene and metagenomics studies showing that these bacterial groups are key plankton components in the Baltic Sea[24–26,33]. As many as 320 out of the 352 BACLs obtained here could not be classified to the species-level, despite the fact that the GTDB also includes species-level clades consisting solely of genomes from uncultured organisms (MAGs and single-amplified genomes). The corresponding numbers for genus- and family-level were 180 and 56. Thus, to our knowledge, the dataset contains substantial novel genomic information. This is also evident by plotting the phylogenetic distances between the BACLs and their nearest neighbors in GTDB, where especially phyla that are represented by a low number of BACL, such as Eisenbacteria, Myxococcaeota and SAR324, display large distances to their nearest GTDB neighbors (Fig. 2).

**Ecological niche distributions**. We used the different metagenomic sample sets to investigate how the BACLs were distributed along various niche gradients in the Baltic Sea ecosystem (Fig. 3). Based on the surface samples from the *Transect 2014* cruise, spanning the salinity gradient from marine to near-freshwater conditions, we derived a salinity niche-parameter for the BACLs by calculating the ratio of their abundances in the high (>14 PSU) vs. low (<6 PSU) salinity samples. Consistent with previous studies[24–26,33], Actinobacteria and Betaproteobacteria where biased toward the lower range of the salinity gradient, while Alpha- and Gammaproteobacteria where biased toward the upper range (Fig. 3b). By taking the ratio between the surface and mid layer samples from the same cruise, we could compare the populations'

relative abundances in sunlit vs. dark conditions (Fig. 3d). As expected, phototrophic Cyanobacteria had a preference for the upper sunlit water layer. In contrast, Planctomycetes, and even more so Crenarcheaota and Thermoplasmataeota, showed a bias towards deeper water layers. Other taxa such as Actinobacteria and Bacteroidetes displayed more variability in their depth preferences, likely reflecting niche-partitioning within these phyla. Finally, we used the data from different filter-size fractions from the *Askö Time Series 2011* to assess the ratio between abundance on >3.0 μm and 0.8–3.0 μm filter fractions (Fig. 3g). Actinobacteria, Alpha- and Gammaproteobacteria were highly underrepresented within the 3.0 μm fraction, consistent with these cells being primarily free-living and rarely particle-associated[34,35]. For Cyanobacteria, BACL annotated as Nostocales and Pseudanabaenales, ie. filamentous cyanobacteria, were enriched on the 3.0 μm filter, consistent with these forming filaments that were captured on the filter, while picocyanobacteria had distinctively lower 3.0 μm/0.8 μm ratios. Bacteroidetes and Planctomycetes displayed large variations, consistent with the fact that some organisms from these groups are known to exist on particles[36,37].

**Predicting niche from genome**. We then proceeded to investigate if the BACLs' distributions along the above described niche gradients could be directly predicted from their genomes. The large number of BACLs allowed us to use a machine learning approach, where we conducted training and predictions on separate sets of BACLs. The encoded genes in each MAG were functionally annotated using eggNOG orthologous groups[38] and a gene (eggNOG) profile was calculated for each BACL based on the mean profile of its MAGs (see Methods). We used various machine learning approaches (ridge regression[39], random forest[40] and gradient boosting[41]) to predict the placement of each BACL along the niche gradients based on its gene profile. For all methods and for all three niche gradients, the gene profile-predicted and actual placements of BACLs were significantly correlated (Spearman rank correlation, $\rho = 0.70$–$0.81$, all $P < 10^{-16}$; Fig. 3c, e, h; Supplementary Table 2).

While the above illustrates that bacterioplankton population distributions can be predicted along specific a priori defined niche gradients, it is reasonable to assume that each population is in fact regulated by a multitude of abiotic and biotic factors. Defining and measuring these factors, such as the availability of specific dissolved organic matter compounds[42] or the presence of specific viruses or predatory protists[43], remains a major challenge in microbial oceanography. These factors will collectively determine a population's abundance in a sample, and thus its abundance profile across multiple samples. Consequently, if two populations display similar abundance profiles across samples they are likely regulated in similar ways and hence likely to share the same ecological niche. Analysing abundance profiles does not require prior knowledge on regulating factors, as long as the samples cover sufficient variation in these, and it allows a quantitative assessment of niche sharing between populations. We retrieved the abundance profile for each BACL over all the metagenome samples (see Methods), and created a low dimensional virtual niche space by running ordination on these profiles (Fig. 4a–d). The first principal coordinates, or dimensions, in this space explain most of the variation in abundance profile and should thus correspond to the highest ecological variation. Among the environmental parameters measured, temperature, oxygen and silicate concentration were the most highly correlated to the first three dimensions, respectively (Fig. 4c, d). However, dimensions of lower rank did not correlate to any of the measured variables, and are presumably driven by other factors. We used machine learning to predict the placement of each BACL in this niche

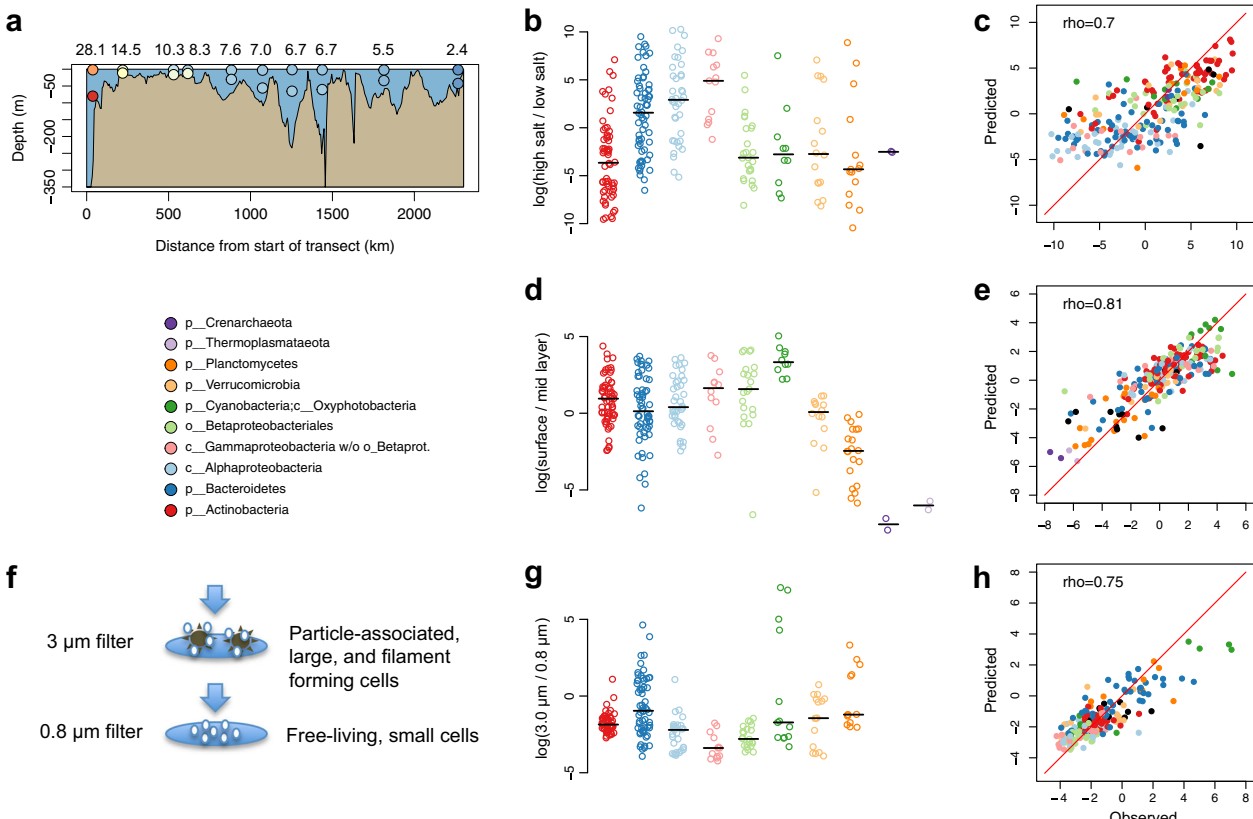

**Fig. 3 Observed and predicted distributions of BACLs along selected niche gradients. a** Side view of *Transect 2014* with surface and mid layer samples indicated by circles, colored according to salinity as in Fig. 1. Numbers above the graph indicate salinity in the surface layer samples. **b** Ratio between abundance in the high and low salinity surface samples of the *Transect 2014* cruise. Values are log ratios of the mean abundances in the 14.5 and 28 PSU and the 2.4 and 5.5 PSU samples. Distributions are plotted separately for each taxon, with median values indicated by horizontal lines. **c** Machine learning predicted vs. observed log ratio between abundance in the high and low salinity samples. **d** Ratio between abundance in surface and abundance in mid layer water samples from the *Transect 2014* cruise. Values are average log ratios for the 10 surface/mid sample pairs. **e** Machine learning predicted vs. observed log ratio between abundance in surface and mid layer samples. **f** Cartoon indicating difference between cells captured on 3 and 0.8 μm filters by sequential filtration. **g** Ratio between abundance on 3.0 μm and abundance on 0.8 μm filters in the *Askö Time Series 2011* sample set. Values are average log ratios for the six 3.0 μm/0.8 μm sample pairs. **h** Machine learning predicted vs. observed log ratio between abundance on 3.0 and 0.8 μm filters. Machine learning predictions performed by gradient boosting using gene (eggNOG) profiles. Low abundance BACLs were excluded from the graphs in **b**, **d**, **g** (see Methods).

space based on its gene profile, again conducting training and predictions on separate BACLs. As for the a priori defined niches, predicted values were significantly correlated to the real values for the first ten principal coordinates of the niche space (Fig. 4e–g and Supplementary Table 2).

**Gene content vs. phylogenetic signal**. Since phylogeny is known to be related to both gene content[44] and abundance distribution[5], it is possible that the machine learning models are merely picking up a phylogenetic signal. Therefore, we also predicted the placement of BACLs in the niche space using phylogenetic information, applying a method based on ancestral state estimation[45]. This method also gave significant correlations to the real values, however with lower correlations for 8 of the first 10 principal coordinates compared to gene-content-based predictions with the machine learning approach that worked best (gradient boosting; Supplementary Table 2). Thus, the gene-based models appear to pick up genetic signals that are directly related to ecology and not only to phylogeny. To further investigate how ecology is reflecting phylogeny as compared to gene content, we correlated pairwise dissimilarity in abundance profile to either pairwise phylogenetic distance or gene profile dissimilarity. A weak but highly significant correlation was found between abundance profile

dissimilarity and phylogenetic distance (Fig. 5a), similar to what was previously observed in a time-series analysis of bacterioplankton[5]. However, this correlation was slightly weaker than between abundance profile dissimilarity and gene profile dissimilarity (Fig. 5b), despite that pairwise dissimilarity in gene profile was highly correlated with phylogenetic distance (Fig. 5c). The stronger correlation between abundance profile and gene content was confirmed by partial correlations, where abundance profile dissimilarity remained correlated with gene content dissimilarity when controlling for phylogenetic distance (partial Mantel test, Spearman $\rho = 0.21$, $P = 10^{-4}$, number of permutations $= 10^4$), while the correlation between abundance profile dissimilarity and phylogenetic distance disappeared when controlling for gene content dissimilarity ($\rho = -0.06$, $P = 1$).

The above gene profile-based niche predictions were conducted using the whole community of BACLs for defining the niche space. We finally performed the same type of analysis, but now generating the virtual niche space and running the machine learning on one taxonomic division at a time, to see if we could resolve more subtle differences in niche based on more subtle differences in gene content. For the clades with most BACLs (Actinobacteria, Bacteroidetes, Alpha- and Gammaproteobacteria) the first three principal coordinates could be predicted fairly well, with mean correlation coefficients between predicted and

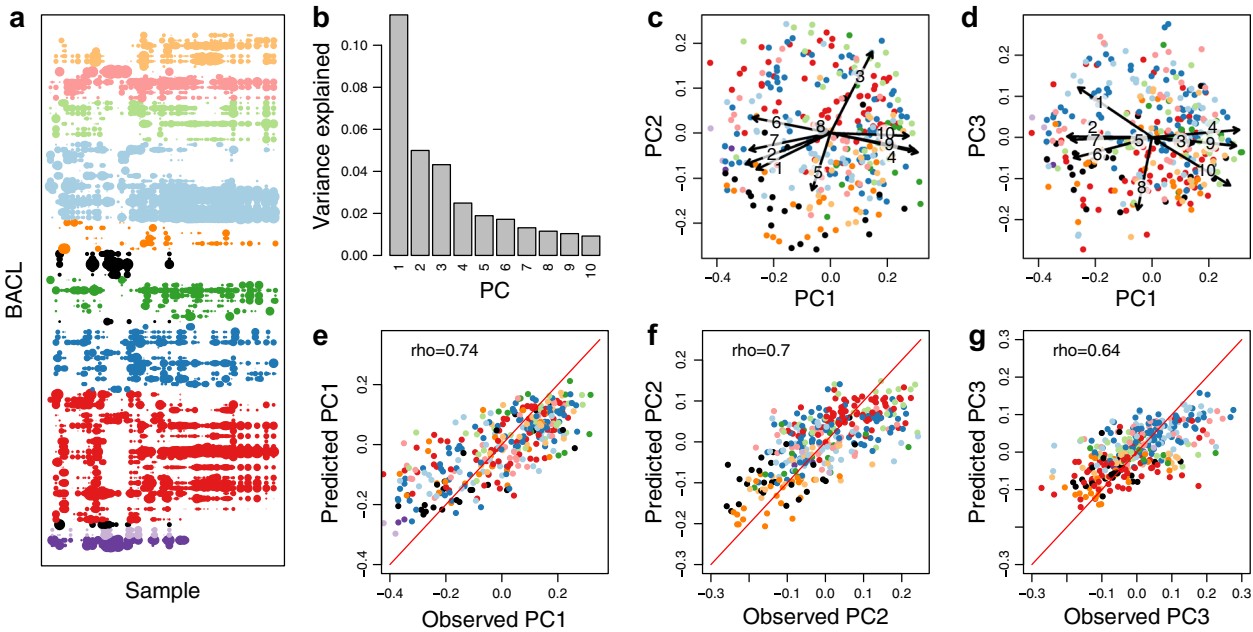

**Fig. 4 Observed and predicted distributions of BACLs along principal axes of abundance variation. a** BACL abundance profiles (one BACL per line; the 99 most abundant BACL shown) across all 124 samples, with dot size proportional to log abundance in the sample, using the same color schema as in Fig. 3 but with additional taxa shown in black. **b**–**d** Principal coordinates analysis of BACL abundance profiles, with **b** displaying proportion of variation explained by the ten first principal coordinates (PC) and **c**, **d** plotting the BACLs along the first three principal coordinates. The arrows indicate relationships between the principal coordinates and measured environmental parameters (see Methods), where the numbers correspond to 1: salinity; 2: depth; 3: oxygen; 4: temperature; 5: filter size; 6: nitrate; 7: phosphate; 8: silicate; 9: chlorophyll a; 10: dissolved organic carbon. **e**–**g** Machine learning predicted (gradient boosting using gene profiles) vs. observed values of principal coordinate scores, with **e** displaying results for PC1, **f** for PC2 and **g** for PC3. Rho-values indicate Spearman rank correlation coefficients between predicted and observed values (all correlations $P < 10^{-16}$). Prediction results for PC1–PC10 using different machine learning algorithms can be found in Supplementary Table 2.

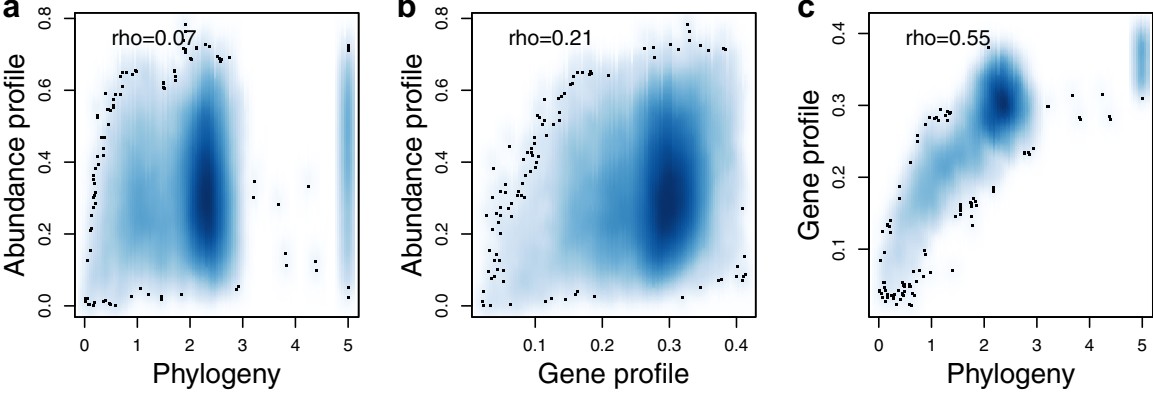

**Fig. 5 Relationships between ecology, phylogeny and gene-content. a** Abundance profile dissimilarity (y-axis) vs. phylogenetic distance (x-axis). **b** Abundance profile dissimilarity (y-axis) vs. gene profile dissimilarity (x-axis). **c** Gene profile dissimilarity (y-axis) vs. phylogenetic distance (x-axis). Rho-values indicate Spearman rank correlation coefficients. All correlations were significant (Mantel test, $P = 10^{-4}$, number of permutations = $10^4$). The background color indicates density of datapoints (BACLs). Individual data points are not shown, except those falling in low density areas (black dots).

real values of 0.61 using gradient boosting (Supplementary Table 3). Again, gene-content-based predictions were generally better than predictions based on phylogenetic information (Supplementary Table 3).

## Discussion

The results presented here demonstrate a strong link between an organism's encoded genes and its ecological niche. Already in the early days of microbial genomics, a relationship between gene content and phylogeny was demonstrated[44] and phylogenetic relatedness has been correlated with ecological relatedness in both macro- and microorganisms[3–5,46–50]. Moreover, genomic

approaches have correlated variation in gene content in natural microbial populations to varying environmental conditions[51–53], and clustering prokaryotes based on what genes they encode has been shown to form groups with shared functional and environmental attributes[54]. However, to our knowledge, our study is the first systematic prediction of ecological niche as manifested in species distributions based solely on genomic information. The placements along the first dimensions in the virtual niche space and along the a priori defined gradients could be estimated with correlation coefficients of ~0.7, meaning that around 50% of the variation along these dimensions could be explained by gene content alone. Since the placement along the first principal

coordinates of the niche space were generally better predicted using gene content than phylogenetic information, our results indicate that gene content is superior to phylogenetic information for predicting ecological niche, highlighting the importance of genomic data for advancing the field of microbial ecology. This was also supported by the direct correlations between abundance profile distances and phylogenetic and gene content distances, respectively. The stronger association between ecology and gene content may appear logical, given that gene content does not strictly follow phylogenetic trajectories due to lateral gene transfer events[55,56]. On the other hand, although the MAGs used for the analysis were estimated to be of rather high quality, the gene content-based models should suffer from some extent of incompleteness and impurities in the genomic information due to shortcomings of the assembly and binning processes. In our analysis we predicted the abundance distributions of species-level genome clusters. As methods for strain-level genome reconstructions develop[57,58] the approach can likely be improved by using more precise information on gene content and abundance distributions of individual strains, since even a single gene can have dramatic effect on niche. Also, genes were grouped in rather broad orthologous groups, that are sometimes functionally heterogeneous. Follow-up studies could address if higher accuracy predictions may be achieved by using more refined gene function definitions, or even genotypic variation. Despite the room for further methodological improvements, our analyses demonstrate a strong link between an organism's gene content and its ecology. The approach developed here may in the future be applicable in environmental management, for example for predicting the abundance distributions of alien species arriving in a new ecosystem. It is also possible that species distribution models (SDM), that today are typically built on environmental data alone[59], can be improved by incorporating genomic information. Whilst we applied the approach to prokaryotes, it should be applicable also for microbial eukaryotes as more genomic information is gathered for these.

## Methods

**Sample retrieval and DNA sequencing**. Samples included within this study are divided into five sample sets named *Askö Time Series 2011*, *Redoxcline 2014*, *Transect 2014*, *LMO Time Series 2013–2014* and *Coastal Transect 2015*(Fig. 1a). Metagenome data for three of these have previously been published: *Askö Time Series 2011*[60], *Redoxcline 2014*[33], *Transect 2014*[33]; and two are new to this publication: *LMO Time Series 2013–2014* and *Coastal Transect 2015*. For the published sample sets, only a brief description of sample retrieval is given here. For detailed descriptions, the reader is directed to the respective publication.

The *Askö Time Series 2011*[60] samples ($n = 24$) were collected on six occasions between 14 June and 30 August in 2011. On each occasion, the samples were sequentially filtered through 200, 3.0, 0.8 and 0.1 μm filters. DNA was sequenced from the 3.0, 0.8 and 0.1 μm filters, as well from the water passing the 0.1 μm filter.

The *Redoxcline 2014*[33] samples ($n = 14$) target the transition between oxic and anoxic water and were collected on three occasions in 2014, from the Gotland Deep on October 18 ($n = 2$) and October 26 ($n = 8$) and from the Boknis Eck[61] station on September 23 ($n = 4$). The October 18 samples were captured on a 0.2 μm filter without pre-filtration while all other samples were filtered either on 3.0 μm filter without pre-filtration ($n = 6$), or on a 0.2 μm filter using 3.0 μm filter for pre-filtration ($n = 6$).

The *Transect 2014*[33] samples ($n = 30$) were collected during a cruise in June 2014. Samples were taken from three different depths, spanning the oxygenated zone, at ten stations covering the horizontal salinity gradient. Samples were captured on a 0.2 μm filter without pre-filtration.

The *LMO Time Series 2013–2014* samples ($n = 22$) were collected from the Linnaeus Microbial Observatory station 10 km east of Öland (Latitude 56.938436, Longitude 17.06204) from January 2013 to December 2014[62]. 10 liter samples from surface water (2 m depth) were collected using a Ruttner sampler and transported to the laboratory in carefully acid rinsed polycarbonate containers. 3–5 liter of seawater were filtered through 0.22 μm filters (Sterivex, Millipore), following pre-filtration through 3.0 μm filters (Poretics polycarbonate, GVS Life Sciences). DNA was extracted using the protocol by Boström et al.[63], as modified by Bunse et al.[64].

The *Coastal Transect 2015* samples ($n = 34$) were collected during a cruise with the R/V Poseidon (Cruise POS488) organised by the Leibniz Institute for Baltic Sea

Research, Warnemünde, in August/September 2015 from stations located closer to the coastline compared to the *Transect 2014* stations. 1 liter samples were collected from surface water (1.7–4.0 m depth) and cells were captured on 0.2 μm filters without pre-filtration. DNA was extracted as earlier described for the *Transect 2014* samples[33].

All sequencing libraries were prepared with the Rubicon ThruPlex kit (Rubicon Genomics, Ann Arbor, Michigan, USA) according to the instructions of the manufacturer and sequenced at the National Genomics Infrastructure (NGI) at Science for Life Laboratory, Stockholm, Sweden, using HiSeq 2500 high-output producing an average of 44 million pair-end read pairs per sample.

**Sequence preprocessing, assembly and quantification**. All samples were pre-processed by the same procedure, removal of low quality bases using cutadapt[65] with parameters "-q 15,15" followed by adapter removal with parameters "-n 3 –minimum-length 31 -a AGATCGGAAGAGCACACGTCTGAACTCCAGTC AC -G ^CGTGTGCTCTTCCGATCT -A AGATCGGAAGAGCGTGTAGGG AAAGAGTGT". These settings ensured that reads shorter than 31 bases after adapter trimming were discarded. Furthermore, the read files were screened for artificial PCR duplicates using FastUniq[66] with default parameters.

After preprocessing, the samples were individually assembled using MEGAHIT[67] version 1.1.2 with the –meta-sensitive option. For each sample, contigs longer than 20 kb were then cut up from the start into non-overlapping parts of 10 kilobases, such that the last piece was between 10 and 20 kilobases long. This was performed using the script "cut_up_fasta.py" from the CONCOCT[29] repository https://github.com/binpro/CONCOCT.

The process continued sample-wise with quantification of each processed assembly file using all read files. The cut-up contigs, as well as all short contigs, were used as input to the index method of Kallisto[28] version 0.43.0. The quantifications were performed using the "quant" method of Kallisto on each of the 124 samples in a cross-wise manner, resulting in $124 \times 124 = 15376$ runs. To transform the estimated counts, which is reported by Kallisto in approximate coverage values, these count values were multiplied by 200 (a simplification, representing the read pair length) and divided by the contig length. This step was performed using the script "kallisto_concoct/input_table.py" from the toolbox repository https://github.com/EnvGen/toolbox (https://doi.org/10.5281/zenodo.1489089).

One of the *Transect 2014* samples (P1994_109) was accidently not assembled and MAGs were not binned from it, but the sample was included in the quantification of contigs of other samples. Hence binning was done on 123 samples but using quantification information from 124 samples.

**Binning and quality screening**. The SpeedUp_Mp branch of CONCOCT was used for binning of the individual samples. Bin assignments by CONCOCT for cut-up contigs were re-evaluated so that all parts of long contigs were placed in the same bin by majority vote. This was done using the script "scripts/concoct/merge_cutup_clustering.py" within the toolbox repository https://github.com/envgen/toolbox (https://doi.org/10.5281/zenodo.1489089). Based on this second bin assignment, all individual bins were extracted as fasta-files, using the original pre-cut-up contigs. To identify prokaryotic Metagenome Assembled Genomes (MAGs), these bins were evaluated using CheckM[30] version 1.0.7. Bins with an estimated completeness of ≥75% and estimated contamination ≤5% were approved and considered prokaryotic MAGs, fulfilling the criteria of being "substantially complete" (≥70%) and having 'low contamination' (≤5%), according to the controlled vocabulary of draft genome quality[30].

**Fragment recruitment**. Proportion of metagenome reads recruited to MAGs was calculated by randomly sampling 1000 forward (R1) reads from each sample and matching against the contigs of all MAGs, including also the LMO 2012 MAGs[22], with BLASTN, using ≥97% identity and alignment length ≥90% of read length as thresholds for counting a read as matching.

**Clustering and taxonomic annotation of MAGs**. Sequence similarity between all MAGs (including those retrieved here and those retrieved in a previous study from station LMO[22]) was estimated using fastANI[68] using the default $k$-mer length of 16. These sequence similarity estimates were used to cluster the MAGs at 96.5% identity level using average-linkage hierarchical clustering using SciPy version 0.17.0. Taxonomic assignment for all prokaryotic MAGs was performed using the classify_wf method of Genome Taxonomy Database Toolkit[32] (GTDB-Tk) using release version 80 of the database and version 0.0.4b1 of the toolkit. Each cluster of prokaryotic MAGs was assigned an identifier *BACLX*, following the nomenclature established in Hugerth et al.[22].

When analysing how BACLs were distributed over niches in the ecosystem and predicting niches, a single MAG was chosen as representative for each MAG cluster. This choice was based on the estimated completeness and contamination levels, where the MAG with highest completeness after subtracting its contamination was chosen. The selected MAGs had a mean estimated completeness and contamination of 92.2% and 2.2%, respectively.

**Evaluation of binning based on internal standard**. Comparisons between the obtained internal standard genome bins and the reference genome (*Thermus thermophilus* str. HB8; accession number GCF_000091545.1) were performed using the dnadiff script from MUMmer version 3.23, comparing to the main reference genome and the two plasmids separately.

**Genome annotations**. Genes were predicted in the MAGs with Prodigal (v.2.6.3), running the program on each MAG separately in default single genome mode. Functional annotation of genes were conducted using eggNOG mapper version 1.0.3[69]. Gene profiles were obtained by counting the number of occurence of each eggNOG with a "@NOG" suffix in each genome. In total 35,593 such unique eggNOGs were found, of which 4115 were COGs. The gene profile of a BACL was calculated by taking the average of the gene profiles of the MAGs in the BACL. Pairwise dissimilarities of gene profiles between BACLs were calculated using Spearman rank correlations, where the gene profile dissimilarity = $(1 - \rho)/2$, and where $\rho$ is the Spearman correlation coefficient.

**Abundance profiles**. The abundance of a MAG in a sample was calculated by taking the average of the Kallisto estimated contig abundances, weighted by the contig lengths, and converted into a coverage per million read-pairs value by dividing by the number of million read-pairs that were mapped from the sample. The abundance profile of the representative MAG for a BACL was used as abundance profile for the BACL (abundance profiles were highly correlated between MAGs within BACLs, average Spearman correlation coefficient = 0.98). Pairwise dissimilarities of abundance profiles between BACLs were calculated using Spearman rank correlations, analogously to how gene profile dissimilarities were calculated. Ordination of abundance profiles was conducted using Principal Coordinates Analysis (PCoA) on the abundance profile dissimilarity matrix using 'Cailliez' correction with the R[70] package ape[71]. To relate the PCoA coordinates to environmental factors (the arrows of Fig. 4c, d), the Spearman correlation coefficients between each BACL abundance profile and each of the measured environmental parameters were first calculated. Next, the Spearman correlation between these correlation coefficients and the BACLs positioning along the PCoA coordinates were calculated. The end-point of the arrow is proportional to the latter correlation: An arrow pointing far to the right indicates that BACLs to the right in the plot are positively correlated with the environmental factor, while those to the left are negatively correlated. An arrow pointing far to the left indicates that BACL to the left in the plot are positively correlated, while those to the right are negatively correlated.

**Phylogenetic distances**. Phylogenetic distances between MAGs were calculated using the R package ape based on the GTDB phylogenetic trees (one for Bacteria and one for Archaea) with MAGs inserted using GTDB-Tk[32] using release version 80 of the database and version 0.0.4b1 of the toolkit. Phylogenetic distances between each bacterial-archaeal pair was set to an arbitrary level of 5 (higher than any of the distances observed within each domain-specific tree). Phylogenetic trees were visualised with GraPhlAn[72].

**Ecological predictions**. In order to lower the risk of miscalculating abundances due to non-specific contig quantifications, BACLs including any MAG with >0.95 ANI to any MAG of another BACL were excluded, leaving 342 BACL for the analysis. All of these were included for the predictions of PCoA coordinate scores (or the subset of these that had the correct taxonomic annotation, when performing taxon-specific predictions). For predicting the a priori defined niches, BACLs among these that displayed low abundances were further removed: When predicting abundance ratio between high and low salinity samples from the *Transect 2014* cruise, only BACLs displaying a highest relative abundance of >0.01 coverage per million read-pairs among these samples were included ($n = 243$). When predicting the average log ratio between the abundance in surface and abundance in mid layer water in the *Transect 2014* cruise, only BACLs displaying a highest coverage of >0.05 coverage per million read-pairs among these 20 samples where included ($n = 246$). When predicting the average log ratio between the abundance on 3.0 μm and abundance on 0.8 μm filters for the *Askö Time Series 2011* sample set, only BACL displaying a highest coverage of >0.01 coverage per million read-pairs among these 12 samples where included ($n = 227$). The same inclusion criteria were used when plotting BACLs along these niche gradients in Fig. 3.

Ecological predictions were conducted using either gene profiles or phylogenetic information. For gene profile-based predictions, gene profiles (calculated as described above) were filtered to only include those eggNOGs that were present in at least 10% of all BACL, resulting in profiles of 3476 eggNOGs of which 2360 were COGs. Gene profile-based predictions were conducted using ridge regression, random forests and gradient boosting. Ridge regressions were performed using the R package glmnet[39] with the alpha parameter set to 0. The hyperparameter lambda was tuned using cross validation within each training set, and the lambda value giving the minimum mean error was used. Random Forest regressions were conducted using the R package randomForest[73], using number of trees set to 2000 (other parameters kept at default values). Gradient boosting regressions were conducted using the R package gbm[74] using a gaussian loss function. The parameter settings for number of trees ('n.trees'), learning rate ('shrinkage'),

maximum depth of each tree ('interaction.depth') and minimum number of observations in the terminal nodes ('n.minobsinnode') were optimised manually based on the success of predicting the scores of the first PCoA coordinate (with all BACL) using different settings. These setting (n.trees = 10000, shrinkage = 0.001, interaction.depth = 2, n.minobsinnode = 1) were subsequently used for all predictions.

Predictions based on phylogenetic information were conducted using the R package picante[45] using ancestral state estimation to infer unknown trait values for taxa based on the values observed in their evolutionary relatives[75,76]. The GTDB trees with inserted MAGs were used for this purpose, by first removing all branches corresponding to other genomes than the BACL representative MAGs.

For ridge regression and gradient boosting we used 10-fold cross-validation between the predicted and observed values. In other words, the set of BACLs were randomly partitioned into ten equally sized subsets. Of the 10 subsets, a single subset was kept as the validation data, and the remaining nine subsets were used as training data. The cross-validation process was then repeated ten times, with each of the ten subsets used once as the validation data. This way, the prediction for each BACL was validated once. For random forests we compared the out-of-bag predictions with the observed values, where the out-of-bag predictions are the predictions based on trees trained on BACLs other than the BACLs under validation. For validations, predicted values were compared with actual values using Spearman rank correlation for all types of predictions.

**Statistics and reproducibility**. Spearman rank correlation was used to evaluate ecological niche predictions and (partial) Mantel test to assess correlations between abundance profile dissimilarity, gene profile dissimilarity and phylogenetic distance.

**Reporting summary**. Further information on research design is available in the Nature Research Reporting Summary linked to this article.

## Data availability
The contigs from the individual samples and the MAG sequences were submitted to ENA hosted by EMBL-EBI under the study accession number PRJEB34883. Note that contigs stemming from the internal standards genome (*Thermus thermophilus*) are included in the contigs for the *Transect 2014* samples. The preprocessed sequencing reads from the *LMO Time Series 2013–2014* and *Coastal Transect 2015* samples were submitted to ENA under the same study accession number (PRJEB34883). The preprocessed sequencing reads from the *Transect 2014* and *Redoxcline 2014* samples were published elsewhere[33] and are available at ENA under the study accession number PRJEB22997. The raw sequencing reads from the *Askö Time Series 2011* were published elsewhere[60] and are available at NCBI under the study accession number SRP077551.

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

## Acknowledgements

This work resulted from the BONUS Blueprint project supported by BONUS (Art 185), funded jointly by the EU and the Swedish Research Council FORMAS, the Federal Ministry of Education and Research (BMBF) and the Danish Council for Independent Research. Funding was also provided through the Swedish governmental strong research programme EcoChange and the Swedish Research Council VR. Computations were performed on resources provided by the Swedish National Infrastructure for Computing (SNIC) through the Uppsala Multidisciplinary Center for Advanced Computational Science (UPPMAX). DNA sequencing was conducted at the Swedish National Genomics Infrastructure (NGI) at Science for Life Laboratory (SciLifeLab) in Stockholm. We are grateful to Warren Kretzschmar for providing advice on machine learning approaches. Open access funding provided by Royal Institute of Technology.

## Author contributions

A.F.A., J.P., M.L., K.J. and L.R. conceived the study. J.P., M.L., K.J. and M.E. coordinated sampling campaigns. Ca.B., Ch.B., S.B. and K.I. conducted sampling and DNA extractions. C.Q. conducted software development. J.A. and A.F.A. conducted analysis. J.A. and A.F.A. wrote the paper with contributions from all authors. All authors read and approved the final version of the paper.

## Competing interests

The authors declare no competing interests.
