## [Peer Review File · Communications Biology]

Reviewers' comments:

Reviewer #1 (Remarks to the Author):

Anelberg, Pinhasi, Andersson et al. perform a relevant study on the prediction of ecological niches from genomic data. This study provides new insights in the field of marine microbial ecology and merits publication after major revision. I list directly beneath a number of issues that need revision.

-please explain it better the rationale. it is weird.

lines 109-110 "Each metagenome sample was assembled and binned separately, but using abundance information from across all samples for the binning."

-Which are the niche gradients? Fig 2 indicates salinity..please clarify
Lines 181-193

0.58 does not look like a reliable correlation value. please check p value.
Line 188- 190

-this part is a bit confusing for the reader.. First, the graph B is not clear. bars , see y axis . also contains PC 1-10 but is not clear if those are the principal coordinates...which are the env parameters?
Lines 205- 215

-" the first systematic prediction of ecological niche as manifested in species distributions based solely on genomic information". well , make sure to go to ncbi and make a proper search..Lines 257-258

EG Biosensors" de Smith et al., 2015 also uses machine learning to predict env parameters from genomic attributes..

see also F Coutinho et al. 2017 Nature Comms. 10.1038/ncomms15955 STOTEN 2019
10.1016/j.scitotenv.2019.04.009

Jensen et al. 2012, 2013 doi: 10.1186/1471-2164-13-S7-S3. 10.12688/f1000research.2
Bayesian prediction of bacterial growth temperature range based on genome sequences.
Bayesian prediction of microbial oxygen requirement.

-Errors related to binning have not been discussed, concerning the estimates of egg coverage
Lines 358-360

-Lines 374-377 needs way better explanation. It is not clear for the reader as written.

-How do the authors make sure relevant eggNOGs have not been removed from the analyzes?? Lines
457-459

Reviewer #3 (Remarks to the Author):

Anelberg et al. present a concept paper on predicting the ecological niche based solely on genomic information. The authors assembled and binned 123 shotgun metagenomes from the Baltic sea with matching metadata, trained machine learning algorithms on part of the dataset and used it to predict the niches of the rest of the dataset.

This manuscript is very well written and has a great flow to it. I edited very little. The methodology used is sound and up to date, and honestly exactly as I would have done it. This paper should be an

example of how to perform this kind of analysis, and that is what makes it important to publish. I especially want to commend the use of a sequencing standard, which is a practice that needs to be much more widely used than it is.

The issue is more the impact and interpretation of the results. I feel like in the end this great dataset and excellent analyses didn't quite deliver. I worry that in the future this paper will be cited for showing a clear correlation between gene content and niche, when it really shows only weak correlations, especially when the phylogenetic signal is removed. I believe the reason is that there aren't really gradients for the main parameters (salinity, depth and size-fraction). From figure 1 it looks to me like salinity is pretty similar except right at the river in the north and right by the Atlantic. You could try this with a finer resolution of salinity and see if you get more of a signal, but I don't require it. It's just an idea for the future. As far as depth and size-fraction, there are only 2 points (in most samples only 1) tested for each as I understand it, so they're hardly gradients. The inconsistency in sampling methods (mainly pre-filtration) also worries me. Was this somehow marked in your metadata? How did you compare samples filtered through a 0.2 with/without prefiltration? Were they considered the same? Obviously, this would have an effect on the abundance profiles. The same goes for the amount of water filtered. Did you use equal amounts of DNA in the library prep?

I also suspect that having time-series data without accounting for seasonality would bias the dataset, because these are simply more samples of the same point that has the same conditions (roughly 1/3 of your samples). When time isn't accounted for in the main analysis, these samples might as well have been collected on the same day. Temperature and NOx were correlated to the second principal component and should have some impact.

Keeping in mind that this is a proof of concept paper, I'm not asking for any re-analyses. I think the authors did a good job with a methodology that should be used by other microbial ecologists. The data is what it is. I would just like to see the issues with the dataset and the results discussed and perhaps emphasized more. We all want to show our best results, but to call a $\rho=0.21$ Spearman correlation coefficient a correlation that can be used to make predictions is a stretch in my opinion.

Another potential drawback is the clustering of MAGs into BACLs and picking only one representative. One could imagine, for example, how an almost identical organism could adapt to higher salinity with addition of a single gene for an osmolite, or with addition of a plasmid, both of which wouldn't show in this analysis at all. I think this point is worth mentioning in a sentence or two.

Specific comments:

Figure 2: I find the observed vs. predicted salinity (fig. 2C) not very well correlated. The ρ may be 0.58 but the plot doesn't look predictive. Was the predicted salinity a continuous variable? This would potentially look better if it was discrete like the measurements. Considering the difference in salinity values between the Y and X axes, predicting that a MAG that peaked at 15ppt would peak below 10 is problematic, especially given the categories used here (1, 10, 20, 30).

As far as size-fraction – as the axes don't have the same range and there's no 1:1 regression it's hard to gage the correlation. Most importantly, the (0,0) point should be somehow highlighted so it would be easier to see how many true vs. false predictions were made.

The observed data presents good clustering by size fraction for most phyla. The Cyanobacterial depth distribution could be easily explained by high/low light clades, especially if you have PAR measurements for the surface vs. mid-layer.

Bacteroidetes and Planctomycetes - did the same MAGs always show up in the same size fraction?

I would add a 1:1 regression to sub-figures c,e,h.

For the machine learning algorithms it would be nice to see a receiver operating characteristic (ROC) curve to validate the model.

MAG abundance: Did you measure breadth of coverage (i.e. how much of the MAG is covered)? If there is a very conserved area in the genome it could, in theory, increase coverage even if the rest of

the genome has very little mapping, which could bias the peak point in which the MAG is found. Please include this information. This is good work so I assume you either already have this information or you could easily get it. It is unlikely to bias your results and would add credit to the analysis.

Figure 3: (A) I would probably move this to supplemental. While it is informative, it's hard to decipher as a small component in a complex figure and would benefit from taking up an entire page. C,D - these figures really demonstrate the issue with the data and can be used to discuss it. PC1 looks very well correlated with salinity, depth, nitrate, phosphate, temperature, chlorophyll and DOC, all of which are known major impactors of aquatic microbial community composition. If all of these together only account for 20% of the variation, then something major is missing. Again, I'm not asking for a re-analysis. I just want to see this discussed.

Figure 4: Please provide more information in the legend. What do the dots represent? What are the blue areas?

Sup. fig. 2: If you used iTOL to make this figure please cite Letunic I and Bork P (2019) Nucleic Acids Res doi: 10.1093/nar/gkz239 Interactive Tree Of Life (iTOL) v4: recent updates and new developments.

Data availability: The accession numbers are missing.

Do you see any indicator species? BACLs that are highly correlated to certain niches? Perhaps for future work this could yield more significant results.

Line 89: missing "of" after "collection"

Line 197: "Consequently, if two populations display similar abundance profiles across samples they are likely regulated in similar ways and hence likely to share the same ecological niche." This phrasing sounds problematic to me. Cyanobacteria and SAR11 share the same niche but are regulated by a different set of factors, especially when it comes to eukaryotic vs. viral predation, but certainly also nutrients.

Line 245: Mean correlation coefficient 0.61: I feel like a box plot would be a more informative representation of this data. I'd like to see not just the mean but the range and preferably raw datapoint plotted on it.

Methods: for the published datasets it would be helpful to add a citation to the papers within the dataset description. I would also add at least the extraction method to the minimal description as it is a great source of bias.

I just want to reiterate here that I appreciate this work. Reviews can be discouraging and disheartening, especially after putting in this much effort. In my opinion, this manuscript should be revised and accepted.

Reply to reviewer's comments

Reviewer #1 (Remarks to the Author):

Anelberg, Pinhasi, Andersson et al. perform a relevant study on the prediction of ecologi niches from genomic data. This study provides new insights in the field of marine microbial ecology and merits publication after major revision. I list directly beneath a number of issues that need revision.

-plese explain it better the rationale. it is weird.

lines 109-110 "Each metagenome sample was assembled and binned separately, but using abundance information from across all samples for the binning."

REPLY: The sentence has been reformulated to "Each metagenome sample was assembled and binned individually, but using abundance information from across all samples for the binning." Hope this makes it clearer (it is common practice today to do binning this way rather than on a co-assembly of all samples).

-Which are the niche gradients? Fig 2 indicates salinity..please clarify
Lines 181-193

REPLY: The niche gradients are: salinity maximum, surface - mid water layer ratio and 3.0 μm - 0.8 μm filter size ratio. The niche gradients are explained in the previous paragraph, as was indicated on line 181 ('above niche gradients'). To clarify this further, we now write 'above described niche gradients' on line 182 in new version. (In response to the comments from another reviewer, the salinity maximum parameter has now been changed to high salinity / low salinity ratio.)

- 0.58 does not look like a reliable correlation value. please check p value.
Line 188- 190

REPLY: The p-values are provided on line 191 (new version) and are $< 10^{-16}$ for all three correlations, hence highly significant.

-this part is a bit confusing for the reader.. First, the graph B is not clear. bars , see y axis . also contains PC 1-10 but is not clar if those are the principal coordinates...which are the env parameters?
Lines 205- 215

REPLY: Fig 3b (now named Fig. 4b) is explained in the figure text: "(b) displaying proportion of variation explained by the ten first principal coordinates (PC)". The ordination was conducted purely based on the abundance profiles of the genomes; the environmental parameters were only compared to the ordination dimensions afterwards. This is hopefully clearer now that we have added the

sentence: “However, dimensions of lower rank did not correlate to any of the measured variables, and are presumably driven by other factors.” (lines 210-212)

- "the first systematic prediction of ecological niche as manifested in species distributions based solely on genomic information". well , make sure to go to ncbi and make a proper search..Lines 257-258

EG Biosensors” de Smith et al., 2015 also uses machine learning to predict env parameters from genomic attributes..

see also F Coutinho et al. 2017 Nature Comms. 10.1038/ncomms15955

STOTEN 2019 10.1016/j.scitotenv.2019.04.009

Jensen et al. 2012, 2013 doi: 10.1186/1471-2164-13-S7-S3.

10.12688/f1000research.2

Bayesian prediction of bacterial growth temperature range based on genome sequences.

Bayesian prediction of microbial oxygen requirement.

REPLY: Smith et al. mBIO 2015 used community profiles to predict environmental parameters, Coutinho et al. Nature Comms. 2017 predicted microbial hosts for viruses, while Coutinho et al. Sci Total Environ. 2019 predicted bacteria, chlorophyll and Vibrio using environmental parameters. The analyses of these are all conceptually very different from what we are doing in this paper, where we predict niches of individual species based on their genomes. Jensen and Ussery F1000Res 2013 and Jensen et al BMC Genomics 2012 do however predict oxygen requirement and growth temperature optimum from the genome, which is related to what we are doing, although they are not doing “prediction of ecological niche as manifested in species distributions” as we are specifically referring to here. In the Introduction we include a number of studies where phenotypic traits are predicted from the genome. We have added the two Jensen reference there (lines 68-69). Thank you for making us aware of these studies.

-Errors related to binning have not been discussed, concerning the estimates of eg coverage

Lines 358-360

REPLY: We now added: “due to shortcomings of assembly and binning” to the end of the sentence “On the other hand, although the MAGs used for the analysis were estimated to be of rather high quality, the gene content-based models should suffer from some extent of incompleteness and impurities in the genomic information” (line 273-274).

-Lines 374-377 needs way better explanation. It is not clear for the reader as written.

REPLY: This has now been reformulated to: “Fragment recruitment. Proportion of metagenome reads recruited to MAGs was calculated by randomly sampling 1,000 forward (R1) reads from each sample and matching against the contigs of all MAGs, including also the LMO 2012 MAGs with BLASTN, using $\geq 97\%$ identity and alignment length $\geq 90\%$ of read length as thresholds for counting a read as matching.” (lines 380-381)

-How do the authors make sure relevant eggNOGs have not been removed from the analyzes?? Lines 457-459

REPLY: This is a good question and the answer is that we can't be 100% sure of this. But our experience is that prediction results are much poorer (and computation time much longer) if we use the unfiltered table of all eggNOGs found in the genomes.

Reviewer #3 (Remarks to the Author):

Alneberg et al. present a concept paper on predicting the ecological niche based solely on genomic information. The authors assembled and binned 123 shotgun metagenomes from the Baltic sea with matching metadata, trained machine learning algorithms on part of the dataset and used it to predict the niches of the rest of the dataset.

This manuscript is very well written and has a great flow to it. I edited very little. The methodology used is sound and up to date, and honestly exactly as I would have done it. This paper should be an example of how to perform this kind of analysis, and that is what makes it important to publish. I especially want to commend the use of a sequencing standard, which is a practice that needs to be much more widely used than it is.

REPLY: Thank you for these very positive comments.

The issue is more the impact and interpretation of the results. I feel like in the end this great dataset and excellent analyses didn't quite deliver. I worry that in the future this paper will be cited for showing a clear correlation between gene content and niche, when it really shows only weak correlations, especially when the phylogenetic signal is removed.

REPLY: We believe the reviewer is referring to the correlation of Fig. 4a here. However, Fig 4a shows the direct correlation between distance in gene content and distance in abundance distribution across samples, weighing all genes equally much when calculating the distance between two genomes. The main message of the paper is that we can predict niche from the gene content, not that there is a strong direct correlation between gene content and niche. In the predictions, the machine-learning algorithm extracts the genes that are important for predicting the specific niche parameter, which we claim works pretty well. For

most niche parameters, predictions work better when basing these on gene content than on phylogenetic information. Fig. 4 (now Fig. 5) mainly serves to show, in a complementary way, that gene content is more related to ecology (i.e. abundance distribution) than phylogenetic information.

I believe the reason is that there aren't really gradients for the main parameters (salinity, depth and size-fraction). From figure 1 it looks to me like salinity is pretty similar except right at the river in the north and right by the Atlantic. You could try this with a finer resolution of salinity and see if you get more of a signal, but I don't require it. It's just an idea for the future.

REPLY: Please see our answer to a related comment below.

As far as depth and size-fraction, there are only 2 points (in most samples only 1) tested for each as I understand it, so they're hardly gradients. The inconsistency in sampling methods (mainly pre-filtration) also worries me. Was this somehow marked in your metadata? How did you compare samples filtered through a 0.2 with/without prefiltration? Were they considered the same? Obviously, this would have an effect on the abundance profiles. The same goes for the amount of water filtered. Did you use equal amounts of DNA in the library prep?

I also suspect that having time-series data without accounting for seasonality would bias the dataset, because these are simply more samples of the same point that has the same conditions (roughly 1/3 of your samples). When time isn't accounted for in the main analysis, these samples might as well have been collected on the same day. Temperature and NOx were correlated to the second principal component and should have some impact.

REPLY: It is true that the dataset as a whole is heterogeneous, which is because it consists of several different datasets. However, when we conducted the predictions of the pre-defined niches (in Fig. 2 (new Fig. 3)) we used only subsets of the data: For the salinity and depth-layer analysis, we used samples from the 2014 Transect cruise (the surface samples for the salinity and the surface and mid-layer samples for the depth-layer analysis). These were all sampled in the same way (captured on 0.2 μ m filters without filtration) during the same cruise. In the depth-layer analysis we do not predict e.g. peak-abundance along an actual depth gradient. As you say, we don't have the appropriate data for that. Instead, we predict the extent of enrichment in surface vs. mid-layer samples. The niche-gradient in this case is the (log) ratio between surface and mid-layer abundance, which is a single value based on the mean of the 10 stations from the 2014 Transect cruise. The third pre-defined niche-gradient that we predict placement along is filter-size ratio. Here we instead used samples from the Askö Time Series, the only dataset that had been serially filtered through different filter sizes. Again we predict a type of enrichment, this time between the largest (3 μ m) and the second largest (0.8 μ m) size fractions, by calculating the mean log ratio between the abundances on these for the 6 time points of this time series.

For the second type of niche-predictions (in Fig. 3 (Fig. 4 in new version)), where we don't predict predefined niches but instead predict where genomes are placed in the 'artificial niche-space' that was created by ordination of the genomes' abundance distributions, we do include all samples. This is because here we want to include as much variation as possible in all types of niches (known and unknown) that affect the populations' abundances. For this analysis mixing samples from different filter sizes, seasons, depths, etc. therefore makes sense, although ideally we would have more equal representation of different conditions. But this is the data that we have at hand.

Keeping in mind that this is a proof of concept paper, I'm not asking for any re-analyses. I think the authors did a good job with a methodology that should be used by other microbial ecologists. The data is what it is. I would just like to see the issues with the dataset and the results discussed and perhaps emphasized more. We all want to show our best results, but to call a $\rho=0.21$ Spearman correlation coefficient a correlation that can be used to make predictions is a stretch in my opinion.

REPLY: Again, this correlation is not from the predictions and not the take-home message of the paper, but only a second way of showing that gene content provides more ecological information than phylogeny. Instead, the actual niche-predictions are the take-home message and these generally have much higher correlations between predicted and actual values (Fig. 3c,e,h & Fig. 4e,f,g (using new Fig numbers)).

Another potential drawback is the clustering of MAGs into BACLs and picking only one representative. One could imagine, for example, how an almost identical organism could adapt to higher salinity with addition of a single gene for an osmolite, or with addition of a plasmid, both of which wouldn't show in this analysis at all. I think this point is worth mentioning in a sentence or two.

REPLY: We fully agree with this, and in the Discussion on lines 274-278 we bring this up "In our analysis we predicted the abundance distributions of species-level genome clusters. As methods for strain-level genome reconstructions develop the approach can likely be improved by using more precise information on gene content and abundance distributions of individual strains". We have now added ", since even a single gene can have dramatic effect on niche." in the end of this sentence.

Specific comments:

Figure 2: I find the observed vs. predicted salinity (fig. 2C) not very well correlated. The ρ may be 0.58 but the plot doesn't look predictive. Was the predicted salinity a continuous variable? This would potentially look better if it was discrete like the measurements. Considering the difference in salinity values between the Y and X axes, predicting that a MAG that peaked at 15ppt would

peak below 10 is problematic, especially given the categories used here (1, 10, 20, 30).

REPLY: Salinity is a continuous parameter, but since the analysis was based on 10 samples with their corresponding salinities, the data points only appear at 10 values along the x-axis. The predictions were regression predictions of a continuous variable. As we wrote in the manuscript, the salinity maximum predictions were the weakest of the three pre-defined niches that we predicted, probably because, as you mention earlier, the data points are not evenly distributed along the salinity gradient. We tried narrowing down the salinity range, as you propose above, by excluding the most saline sample (where only 3 BACL peak), but that only increased the prediction accuracy slightly. Instead we now calculated the log ratio between the mean abundance in the two most saline (>14 PSU) and the two least saline (<6 PSU) samples, and predicted this log ratio (new Fig. 3). This parameter is more evenly distributed and the predictions of it are considerably better than of the previous salinity maximum.

As far as size-fraction – as the axes don't have the same range and there's no 1:1 regression it's hard to gage the correlation. Most importantly, the (0,0) point should be somehow highlighted so it would be easier to see how many true vs. false predictions were made.

REPLY: We have now adjusted the axes to have the same range. However, there is nothing really special with the 0,0 point here and there are no true or false predictions, since the predictions are not categorical.

The observed data presents good clustering by size fraction for most phyla. The Cyanobacterial depth distribution could be easily explained by high/low light clades, especially if you have PAR measurements for the surface vs. mid-layer. Bacteroidetes and Planctomycetes - did the same MAGs always show up in the same size fraction?

REPLY: The cyanobacteria are all pretty much clustered together in the depth distribution, but not in the size fractions, where the filamentous ones are biased toward the larger size fraction (as mentioned in the manuscript). Yes, there is a high concordance in the size fraction ratios between different samples. The analysis in the manuscript were based on averages of the 6 samples, but if we calculate the ratios for one sample at a time and correlate between each pair of samples, the mean correlation between a pair of samples is $r = 0.80$ considering all BACL and $r = 0.76$ considering only the Bacteroidetes + Planctomycetes.

I would add a 1:1 regression to sub-figures c,e,h.

REPLY: We have now added 1:1 lines to the graphs.

For the machine learning algorithms it would be nice to see a receiver operating characteristic (ROC) curve to validate the model.

REPLY: Since we don't conduct classification but rather regression a ROC curve is not applicable.

MAG abundance: Did you measure breadth of coverage (i.e. how much of the MAG is covered)? If there is a very conserved area in the genome it could, in theory, increase coverage even if the rest of the genome has very little mapping, which could bias the peak point in which the MAG is found. Please include this information. This is good work so I assume you either already have this information or you could easily get it. It is unlikely to bias your results and would add credit to the analysis.

REPLY: We don't calculate breadth of coverage and it is not clear to us how this information would be used here. But this is to some extent related to strain variation, which we address in the Discussion.

Figure 3: (A) I would probably move this to supplemental. While it is informative, it's hard to decipher as a small component in a complex figure and would benefit from taking up an entire page.

REPLY: We include this mainly for pedagogical purpose, to illustrate that we go from abundance distributions (A) to ordination space (B-D) and prefer keeping it in its current form.

C,D - these figures really demonstrate the issue with the data and can be used to discuss it. PC1 looks very well correlated with salinity, depth, nitrate, phosphate, temperature, chlorophyll and DOC, all of which are known major impactors of aquatic microbial community composition. If all of these together only account for 20% of the variation, then something major is missing. Again, I'm not asking for a re-analysis. I just want to see this discussed.

REPLY: Very good point. We have now added a sentence to lines 210-212 "However, dimensions of lower rank did not correlate to any of the measured variables, and are presumably driven by other factors."

Figure 4: Please provide more information in the legend. What do the dots represent? What are the blue areas?

REPLY: The following has now been added to the legend: "The background color indicates density of datapoints (BACLs). Individual datapoints are not shown, except those falling in low density areas (black dots)."

Sup. fig. 2: If you used IToL to make this figure please cite Letunic I and Bork P

(2019) Nucleic Acids Res doi: 10.1093/nar/gkz239 Interactive Tree Of Life (iTOL) v4: recent updates and new developments.

REPLY: The trees were drawn with GraPhlAn. This information has now been added on line 440. We also note that the text to Supplementary Figure 2 had been lost during conversion to pdf. This has been fixed in the new version.

Data availability: The accession numbers are missing.

REPLY: We are in the process of submitting the MAG sequences to ENA. This takes a long time but we will provide them before the manuscript is accepted.

Do you see any indicator species? BACLs that are highly correlated to certain niches? Perhaps for future work this could yield more significant results.

REPLY: This has not been the focus of this paper, but is certainly interesting for future work.

Line 89: missing "of" after "collection"

REPLY: Now fixed.

Line 197: "Consequently, if two populations display similar abundance profiles across samples they are likely regulated in similar ways and hence likely to share the same ecological niche." This phrasing sounds problematic to me. Cyanobacteria and SAR11 share the same niche but are regulated by a different set of factors, especially when it comes to eukaryotic vs. viral predation, but certainly also nutrients.

REPLY: If they are regulated by different factors they are not entirely sharing the same niche.

Line 245: Mean correlation coefficient 0.61: I feel like a box plot would be a more informative representation of this data. I'd like to see not just the mean but the range and preferably raw datapoint plotted on it.

REPLY: Supplementary Table 4 gives the 16 correlation coefficients that this mean value is calculated upon.

Methods: for the published datasets it would be helpful to add a citation to the papers within the dataset description. I would also add at least the extraction method to the minimal description as it is a great source of bias.

REPLY: The two papers from where the previously published datasets derive are already cited in the first paragraph in 'Sample retrieval and DNA sequencing' in Methods, but citations have now been added to the other paragraphs of this

section as well. We noted that one of the papers was cited as two different papers; this has now been fixed. Since the DNA extraction methods are rather extensive we prefer to cite the original papers for these.

I just want to reiterate here that I appreciate this work. Reviews can be discouraging and disheartening, especially after putting in this much effort. In my opinion, this manuscript should be revised and accepted.

REPLY: Thank you for the constructive and encouraging comments, and for appreciating how much work has been put into this study!

Reviewer #2

The work by Alneberg et al. describes a monumental amount of data and work spanning multiple cruises and sampling expeditions. In their manuscript they describe the generation of almost 2000 MAGs (metagenomically assembled genomes) from the Baltic sea. Subsequently they go into using several machine learning approaches where they use the gene content information and abundance profile of their genomic clusters to place them into a 1) environmental niche space and 2) virtual niche space derived from MAG abundance. They claim that the ability to place these represents a testament of ecological signals ingrained into the genomes of prokaryotes.

Comments:

It is a well thought work with very interesting results, which are novel and potentially provocative I think it will have a good impact in the field. Additionally, it is gifting the communality with hundreds of new reference genomes and metagenomic data for future research.

REPLY: Thank you for these positive comments.

Data for figure 1b. Is there a reason to use blast when you have data from kallisto?, (The authors have the abundance and recruitment numbers as they are an input for CONCOCT and were calculated with Kallisto, I wonder if there is a reason to use blast? How do these numbers change if you use the kallisto/recruitment data.

REPLY: We could probably have calculated these from Kallisto but decided to use BLAST mainly because we wanted to be able to control the sequence identity and alignment length.

Abundance profiles. How does Kallisto deal with equally good alignments?

REPLY: Kallisto uses an EM algorithm to estimate the abundances of the contigs based on its so-called pseudoalignments. If a read matches two contigs equally

well, that read will in the end contribute to the counts of the two contigs proportional to what the algorithm in the end estimates these contigs' abundances to.

Clustering of MAGS. Why was 96.5 Chosen? Did the authors performed a sensibility analysis and it wasn't shown? Interestingly, it is slightly different to that 97% used for their blast recruitment.

REPLY: It was based on the distribution of ANI between MAGs (see Fig. 1c). 96.5% is at the edge of the 'hump' to the right in the graph. For the BLAST recruitments we could have used the same but didn't, simply because in the script that we used we set the identity threshold as an integer. If we would have used 96.5% in the BLAST recruitments we may have gotten marginally more recruitment, but certainly not less.

Would this level of clustering (or type of clustering, ANI) join into a cluster MAGS that are 99% identical over most of the genome, except for a few Islands? I think this could be an interesting point since sometimes Islands encode niche specific genes (see *Prochlorococcus* for example). The authors use one of the MAGs as the representative of the cluster for a big portion of the analysis, but this ignores genomic islands... I think it could be interesting to see this addressed in the manuscript.

REPLY: Yes, such MAGs would be joined. This is related to the Discussion on line 275-276 where we now say: "As methods for strain-level genome reconstructions develop the approach can likely be improved by using more precise information on gene content and abundance distributions of individual strains, since even a single gene can have dramatic effect on niche."

Training sets for machine learning. How where the sets of clusters of MAGS used for training and validation (analyses) chosen? This is important since all the attempts to disentangle ecology/genomic signals from phylogeny seem to be done a posteriori.

REPLY: For the gene-content based models, random (non-overlapping) sets of BACL were used for the training and validation. For the phylogeny-based predictions, the niche-value for one BACL a time was predicted using the niche-values of all the other BACL (that were also in the tree). In both the phylogeny and gene-content based modeling one MAG was chosen to represent each BACL. Hope this answered your question.

REVIEWERS' COMMENTS:

Reviewer #1 (Remarks to the Author):

The authors have corrected all the points raised in the previous round of revision and included additional info in their revised paper. I have no further comments and accept this paper as such. Congratulations.

Reviewer #2 (Remarks to the Author):

The authors have provided thoughtful and detailed responses to the comments. I have no further comments other than:

Can the authors include a table where they describe which clusters were used for model training and which clusters were used to test the predictive power?

I think the point would be better made if you show that some (or hopefully many) of the clusters appear only on one set (either training or test); in that case really the gene content is predicting the ecology and not the phylogenetic relatedness.

The community looks forward to seeing all those fantastic metagenomic data.

Reviewer #3 (Remarks to the Author):

I am happy to say that the authors addressed all of my concerns to my satisfaction. I believe the revised manuscript is ready for publication.

Replies to reviewers' comments

Reviewer #1 (Remarks to the Author):

The authors have corrected all the points raised in the previous round of revision and included additional info in their revised paper. I have no further comments and accept this paper as such. Congratulations.

Reviewer #2 (Remarks to the Author):

The authors have provided thoughtful and detailed responses to the comments. I have no further comments other than:

Can the authors include a table where they describe which clusters were used for model training and which clusters were used to test the predictive power?

I think the point would be better made if you show that some (or hopefully many) of the clusters appear only on one set (either training or test); in that case really the gene content is predicting the ecology and not the phylogenetic relatedness.

The community looks forward to seeing all those fantastic metagenomic data.

REPLY: All clusters were used for model training and for testing, but never at the same time. Therefore there is no need to include such a table. We have clarified the procedure in the Methods on lines 483-492 and rephrased lines 184-186 in Results to make it even clearer that separate BACLs were used in training and testing.

Reviewer #3 (Remarks to the Author):

I am happy to say that the authors addressed all of my concerns to my satisfaction. I believe the revised manuscript is ready for publication.